# VideoUntier: Language-guided Video Feature Disentanglement

## Abstract

Most of existing text-video retrieval works learn features comprehensively representing complicated video contents. This leads to the difficulty of textual-visual feature alignment, because text queries convey more concise cues like certain objects and events the user desires to retrieve. To pursue a more compact video representation and accurate textual-visual feature matching, this paper introduces a novel VideoUntier to disentangle video features. VideoUntier first generates 'object' and 'event' tokens from query texts. It subsequently spots and merges visual tokens related to concepts in the query. In other words, we use 'object' and 'event' tokens to represent cues of query, which therefore supervise the disentanglement and extraction of meaningful visual features from videos. VideoUntier finally leads to compact visual tokens explicitly depicting query objects and events. Extensive experiments on three widely-used datasets demonstrate the promising performance and domain generalization capability of our method. For instance, our method shows better efficiency and consistently outperforms many recent works like ProST on three datasets. We hope to inspire future work for collaborative cross-modal learning with certain modality as guidance.

## 1 Introduction

With the rapid development of short video platforms, learning spatio-temporal representations has gained significant interest as a fundamental component of video understanding. The complexity of video contents poses challenges in video representation learning. A common practice is extracting raw video features comprehensively representing complicated video contents. This contradicts the fact vision tasks commonly rely on a particular aspect of video contents. For instance, human action recognition concentrates on detailed motion cues of the human body. While for VideoQA, the model needs to learn semantics and relationships among objects, rather than depicting detailed low-level cues. This leads to a fundamental challenge in video representation learning: it is difficult to decide the most valuable visual cues for a specific task before the vision model is extensively trained.

This challenge has hindered the efficiency of Text-Video Retrieval (TVR) task. "A picture is worth a thousand words", indicating that text queries and videos exhibit distinct information densities. In contrast to videos that often include considerable query-irrelevant cues, textual queries tend to be more concise, focusing solely on specific objects and events the user wants to retrieve. Such differences would make extracted video features easily suffer from noises depicting query-irrelevant visual cues. During the retrieval procedure, these noisy features may depress the saliency of visual features related to the query, resulting in a misalignment between the video and textual modalities. For example, as shown in fig:teaser, the video includes many concepts like 'teacher', 'students', and 'bookshelf' but the text query is only concerned with 'teacher' and his actions.

This work aims to alleviate the misalignment between video and text query features from a different perspective. As the video-text retrieval task provides text queries, we aim to extract more compact and valuable video features by referring to text queries, rather than solely relying on end-to-end model training. We propose the VideoUntier to disentangle video features into various types of tokens, with the query text as guidance. VideoUntier first generates 'object' and 'event' tokens from query texts. It subsequently spots and merges visual tokens related to concepts in the query. In other words, we use 'object' and 'event' tokens to represent cues of query, which therefore supervise the disentanglement

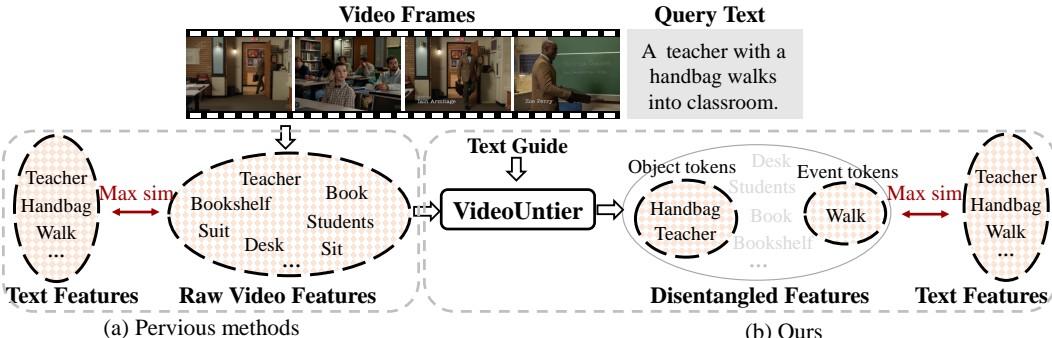

Figure 1: (a) Previous works directly pull close the distance between raw video features contains considerable noises depicting query-irrelevant visual cues, leading to difficulties in aligning multi-modality features. (b) VideoUntier generates video features with guidance from object and event tokens of the query text, hence effectively guarantees a more robust and accurate feature alignment.

and extraction of meaningful visual features from videos. VideoUntier finally leads to compact visual tokens explicitly depicting query objects and events.

More specifically, VideoUntier disentangles video content into global, object, and event levels of features for multi-grained feature alignment. Global features are directly generated from the pretrained backbone such as CLIP (Radford et al., 2021), aiming at the efficient global level alignment. Object and event-level features are extracted by the guidance of object and event tokens from text queries. VideoUntier comprises two main components. The first is the Part-of-Speech-based Token Generator (PTG) module to parse text queries into noun and verb features. This module enhances these word semantics through contextual cross-attention within sentences, leading to the object and event-related tokens. The second component, the Language-guided Progressive Vision Merging (LPVM) module, proceeds to extract object features and event features from the video. It starts by merging features within individual frames to identify objects, then facilitates interactions among these object features over the temporal dimension to capture dynamic actions and inter-object interactions.

Experimental results on the MSRVTT (Xu et al., 2016), MSVD (Wu et al., 2017), and DiDeMo (Anne Hendricks et al., 2017) show our VideoUntier exhibits promising accuracy and efficiency. Compared to recent fine-grained and multi-granularity alignment works, VideoUntier consistently outperforms them on these datasets. VideoUntier also boosts the feature robustness of raw features learned by the backbone network, which is demonstrated by a notable enhancement in domain generalization experiments. Visualization also illustrates that VideoUntier effectively identifies visual features related to objects and events in the query. We thus conclude that VideoUntier achieves a robust and accurate alignment between textual-visual features.

Existing works on video-text retrieval can be categorized into three categories, based on the granularity of their learnt video features, *i.e.*, global, frame-level, and patch-level features, respectively. Learning more detailed cues like patch-level features generally boosts the retrieval performance, but sacrifices the feature compactness and efficiency. To the best of our knowledge, this work is an original effort in learning object and event features from videos with guidance from text queries in TVR. It shows promising performance in aspects of both retrieval efficiency and accuracy. The promising performance of this work shows that, the text guidance effectively alleviates the difficulties in meaningful video feature learning and video-text feature alignment. Its success may inspire future work on text-guided video feature learning.

## 2 RELATED WORK

Existing cross-modality works between text and video can be roughly categorized into three categories based on the granularity of the learnt alignment, i.e., global, frame-level, and patch-level features.

**Global alignment.** Most early works follow the global matching framework. They involve mapping text and video into a shared latent space, enabling direct calculation of similarity between them. JPoSE (Wray et al., 2019) decomposes text features into the form of a verb plus a noun, but it cannot handle complex sentence structures. Recently, the emergence of large-scale vision-language

pre-trained models, like CLIP (Radford et al., 2021), has remarkably advanced the state-of-the-art performances of cross-modal tasks. Researchers propose to transfer knowledge from cross-modal pre-trained models to spatiotemporal tasks. Among these, ClipBERT (Lei et al., 2021) end-to-end fine-tunes the pretrained model with sparse frame sampling. Frozen (Bain et al., 2021) applies a curriculum learning approach to joint image and video training. CLIP4Clip (Luo et al., 2022) explores various similarity calculation methods and post-pretraining to improve both zero-shot and fine-tuned performance. Centerclip (Zhao et al., 2022) performs segment-level clustering to cut down on token redundancy and computational costs. Nevertheless, the accuracy of global similarity in aligning video-text pairs is limited due to data distribution disparities and the simplicity of vision-text interaction by the dot product of single features.

**Frame-level alignment.** Frame-level alignment represents video frames as feature vectors, which are then used to compute similarity with sentences. X-Pool (Gorti et al., 2022) utilizes cross-modal attention to align text with its semantically closest video frames. Huang et al. (Huang et al., 2023) propose to cooperatively tune video and text prompts for improving the adaptation efficiency. HBI (Jin et al., 2023a) formulates a cooperative game to supervise the alignment between video frames and text words. However, this method, while efficient, grasps only frame-level visual cues, such as scenes and collective actions (Ma et al., 2022), thereby ignoring the intrinsic video-text relations which might diminish the accuracy of retrieval.

**Patch-level alignment.** Patch-level alignment facilitates a direct correspondence between image patches and word tokens, excelling in capturing the subtle object details for precise retrieval. To enhance the learning of local associations, Ge et al. (Ge et al., 2022) propose a masking and recovery pretext task. Additionally, TMVM (Lin et al., 2022) and ProST (Li et al., 2023) propose to aggregate key objects and actions in videos into several prototypes for many-to-many correspondence calculations with sentence features. UCOFIA (Wang et al., 2023) takes a unified approach to cross-modal correspondence, considering interactions at video-sentence, frame-sentence, and patch-word levels. However, these works do not account for the richness of objects and events contained in the video. A video clip often encompasses more information than what is described in its text query. This discrepancy can lead to inconsistent semantic granularity between fine-grained video features and text words, leading to the potential obfuscation of crucial matches by irrelevant background distractions.

**Differences with previous works.** In contrast to prior works using nouns and verbs, our approach explicitly utilizes textual information to bridge the information density gap between text and video modalities. Our differences can be summarized as follows: 1) Different motivations: Our VideoUniter is designed to address the information gap between textual and video modalities by using text as a guide for video feature extraction. 2) Different methodologies: VideoUniter leverages the semantic understanding capabilities of the pretrained CLIP model to guide the extraction of visual features associated with objects and events, without using additional datasets and pretrained models. Our approach avoids the computational expenditure on irrelevant background noise. Incorporating both global and fine-grained alignment enhances the efficiency of both training and inference.

## 3 METHOD

### 3.1 FORMULATION

Text-Video Retrieval (TVR) aims to develop the similarity $S$ within all text-video pairs to measure the relations across modalities, and thus choose the most matched pairs. The matched text-video pairs should be closely aligned and the mismatched pairs should be away from each other. Formally, it can be written as $S_+ > S_-$, which denotes similarities between the positive and negative pairs, separately.

Given a text-video pair $(t, v)$ for similarity $S$ calculation, VideoUntier embeds the input into text features $T$ and video features $V$ through the backbone network. It then disentangles these features into different categories. We use three categories of features to represent the video content, i.e., global, object, and event-level features. The global-level represents the overall content, such as text context and video scenes. Object-level feature detailed semantics of objects in the video. Event-level features describe actions and interactions among objects in the video. We use tokens as the uniform representation of three levels of features, i.e., a single global token, alongside multiple object tokens

and event tokens to capture the rich fine-grained semantics, which are denoted as $\{t^g, \{t^o\}, \{t^e\}\}$ and $\{v^g, \{v^o\}, \{v^e\}\}$ for text and video features, respectively.

Videos inherently contain more complicated contents than text queries. This leads to video-text pairs being partially matched. Video contents that are absent in text can disturb the similarity computation between text and video features, and also potentially lead to network overfitting to these irrelevant details during training. To address this, we leverage text-based information to guide the extraction of video semantics. Firstly, we introduce the PoS-based Token Generator (PTG) module to disentangle essential object tokens $\{t^o\}$ and event tokens $\{t^e\}$ from text embeddings $T$ based on raw textual description $t$. This module is denoted as:

$$\{\{t^o\}, \{t^e\}\} = \text{PTG}(T, t). \tag{1}$$

Guided by the object and event tokens from the text query, we then extract features from the video that are relevant to the query. This approach helps to eliminate irrelevant noise and ensures video features consistent in granularity with the text content. Specifically, we develop a Language-guided Progressive Vision Merging (LPVM) module to extract critical information for the VTR task from video, which can be formulated as:

$$\{\{v^o\}, \{v^e\}\} = \text{LPVM}(V, \{\{t^o\}, \{t^e\}\}). \tag{2}$$

Global-level, object-level, and event-level features are jointly used for the computation of multi-grained similarities between texts and videos. The yielded multi-grained similarity score $S$ can be written as:

$$S = S^g + S^o + S^e, \tag{3}$$

where $S^g, S^o, S^e$ denote global similarity, object similarity and event similarity, respectively. The pipeline of VideoUntier is illustrated in Figure 2 (a).

### 3.2 Global Feature Extraction

We use CLIP (Radford et al., 2021) as the backbone to embed the input text-video pair $(t, v)$ into text features $T$ and video features $V$, for a fair comparison with recent methods (Li et al., 2023; Jin et al., 2023a; Luo et al., 2022). For each query text $t$, CLIP includes [SOT] and [EOT] tokens to mark the start and end of the text. Output words sequence are expressed as $T = [T[\text{SOT}], T[1], \ldots, T[N_t], T[\text{EOT}]] \in \mathbb{R}^{(N_t+2) \times d}$, where $N_t$ is the number of words, $d$ is the number of dimensions, $T[\text{EOT}]$ captures the overall textual semantic, which can be used as our global feature:

$$t^g = T[\text{EOT}]. \tag{4}$$

Input video (or video clip) $v$ is composed of $N_f$ sampled frames and each frame has $N_p$ local patches. Formally, each sampled frame is embedded into sequential features, i.e., $v_j = [v_j[\text{CLS}], v_j[1], v_j[2], \ldots, v_j[N_p]] \in \mathbb{R}^{(N_p+1) \times d}$, where $v_j[\text{CLS}]$ denotes the global frame token [CLS] for the $j$-th frame. The features of all frames are concatenated to form the complete video feature $V = [v_1, v_2, \ldots, v_{N_f}] \in \mathbb{R}^{N_f \times (N_p+1) \times d}$. Following previous works (Luo et al., 2022; Jin et al., 2023a), we process all frame tokens [CLS] through Transformer Encoder (Vaswani et al., 2017) for temporal interaction and then apply mean-pooling to obtain the global video feature $g^v$. This process is formulated as:

$$\tilde{v}_j^g = \text{Transformer-Enc}(v_j[\text{CLS}] + p_j), \tag{5}$$

where $p_j$ is position embedding for the j-th frame, then

$$v^g = \text{Mean-Pooling}(\tilde{v}_1^g, \tilde{v}_2^g, \ldots, \tilde{v}_{N_f}^g). \tag{6}$$

### 3.3 Part-of-Speech-based Token Generator

This section is dedicated to generating text objects $\{v^o\}$ and events $\{v^e\}$ based on the raw textual description $t$ and text embeddings $T$, employing part-of-speech (PoS) tagging. Initially, utilizing the Stanford PoS tagger (Manning et al., 2014), we assign tags such as 'nouns' and 'verbs' to words within the sentences. Subsequently, we extract features corresponding to the indices of nouns and

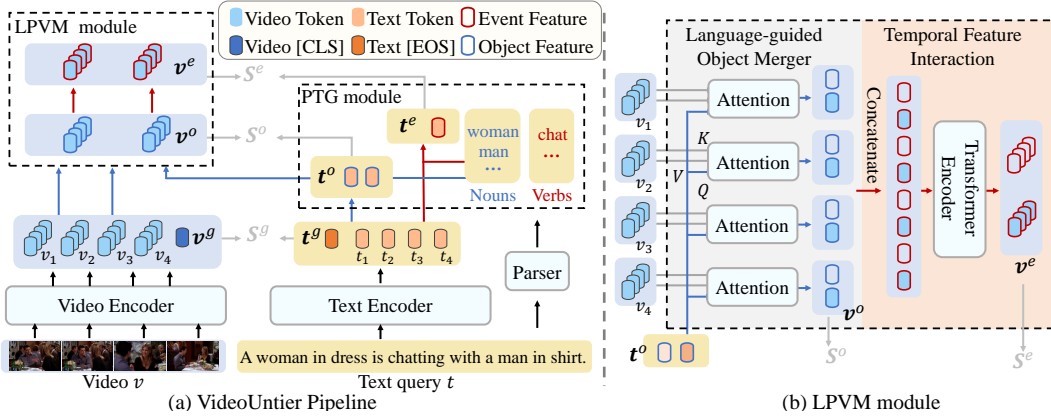

Figure 2: (a) Pipeline of VideoUntier. It comprises two main modules. PTG module disentangles object $t^o$ and event $t^e$ tokens from text. (b) LPVM module extracts video object tokens $v^o$ and event tokens $v^e$ with the guidance of the texts. $v_i$ denotes the token sequence corresponding to $i$-th frame.

verbs from the text embeddings $T$. These extracted features serve as the initial object and event features, defined as:

$$\{\tilde{t}^o\} = \{T_i \mid t_i \text{ is a Noun}\},$$
$$\{\tilde{t}^e\} = \{T_i \mid t_i \text{ is a Verb}\}. \tag{7}$$

For instance, consider the sentence "A woman in a dress is chatting with a man in a suit." We extract nouns {"woman", "dress", "man", and "suit"}, along with the verb {"chatting"}. Then, we gather features corresponding to these words to initialize $\{\tilde{t}^o\}$ and $\{\tilde{t}^e\}$. To facilitate parallel batch computation, we set the number of extracted nouns and verbs as $\left|\{\tilde{t}^o\}\right| = N_{noun}$ and $\left|\{\tilde{t}^e\}\right| = N_{verb}$, respectively. If a sentence contains fewer nouns or verbs than $N_{noun}$ and $N_{verb}$, we use other words from the sentence for padding.

The context of words is crucial for understanding their semantic meanings. For example, to accurately grasp the semantics of "dress" in "woman in dress" or "suit" in "man in suit," it's essential to consider the phrase context. To this end, we aggregate text context by conducting cross-attention (Vaswani et al., 2017) between the extracted word features and the sentence embeddings. Specifically, we use word features $\tilde{t}^o$ and $\tilde{t}^e$ as queries, and text features $T$ as keys and values for attention computation. We incorporate residual connections to ensure the training stability of the network. This process can be formulated as:

$$\{t^o\} = \text{Attention}(\{\tilde{t}^o\}, T, T),$$
$$\{t^e\} = \text{Attention}(\{\tilde{t}^e\}, T, T). \tag{8}$$

The discrete 'noun' and 'verb' tokens are thus transformed to 'object' and 'event' tokens, facilitating fine-grained alignment with the video.

In the case of an incomplete caption with a missing verb or noun, we prioritize words describing certain video content for padding. The priority order for padding is established as follows: noun = verb >adj. >adv. >prep. >conj. >others. It is noteworthy that not only nouns and verbs can capture the semantic information of a sentence. Utilizing words other than nouns and verbs for padding can also reflect the sentence's semantics. For instance, in the sentence "a happy boy is running on the ground", words like "happy" and "on" may not directly correspond to specific objects or actions. However, within the context of text encoding, these words acquire semantics from adjacent words "boy" and "ground," allowing them to capture valuable semantic or syntactic information.

### 3.4 LANGUAGE-GUIDED PROGRESSIVE VISION MERGING

Guided by the information extracted from the text, this section progressively identifies and extracts relevant object and event information from videos as illustrated in Figure 2 (b). This approach ensures consistency in the content of fine-grained features across different modalities.

**Language-guided Object Merger.** Given a video feature composed of multiple frames, we start by extracting object information within each frame. Utilizing the object tokens $\{t^o\}$ extracted from the text as queries, and the patch embeddings $V$ as both key and value, we compute cross-attention (Vaswani et al., 2017), which can be written as:

$$\{\boldsymbol{v}^o\} = \text{Attention}(\{\boldsymbol{t}^o\}, V, V). \tag{9}$$

It's worth noting that the merging process is conducted separately for each frame. As a result, the number of the object features in the video is determined by both the number of frames and the count of object tokens in the text, i.e., $|\{\boldsymbol{v}^o\}| = N_f \times N_{noun}$, where $N_f$ denotes the number of frames, and $N_{noun}$ is the number of generated text object tokens. This operation queries each frame feature, aggregating information relevant to the objects in the query into a unified representation.

**Temporal Feature Interaction.** The extraction of video object features is conducted within individual frames, and the backbone does not consider fine-grained temporal interaction during embedding. As a result, the extracted object information $\{v^o\}$ remains static, lacking the ability to capture the object actions and interactions across time. To effectively leverage the temporal information in videos, we conduct a temporal feature interaction for static features to capture temporal semantics. Specifically, we initialize event features using static object characteristics. At this stage, these event features correspond to static text objects, but do not include information over temporal sequence. These event features are then processed through a Transformer Encoder to enable temporal interactions. This step models the dynamic actions and inter-object interactions, effectively capturing the events happening in videos. This operation can be written as:

$$\{\boldsymbol{v}^e\} = \text{Transformer-Enc}(\{\boldsymbol{v}^o + p\}), \tag{10}$$

where $p$ is the positional embedding in the temporal dimension. Note that the sequence length of the event features here is equal to the length of visual object features, not corresponding to the length of text event features. Therefore, we will not match them one-to-one. Instead, we will conduct a top-$K$ matching according to Eq. 13.

**Discussion.** Nouns are often accompanied by descriptive adjectives, such as "happy kids", and verbs are frequently paired with corresponding adverbs, like "shout angrily". When conducting text encoding using CLIP, these related components are positioned close to each other in the embedding space, reflecting their semantic connections. To leverage this inherent relationship between nouns and adjectives, as well as between verbs and adverbs, we have designed Eq. 8. This formulation enables the features of nouns and verbs to integrate the contextual information from these adjectives and adverbs through attention mechanisms. Given this high degree of correlation, preserving both the adjectives (alongside the nouns they describe) and the adverbs (alongside the corresponding verbs) would lead to representational and computational redundancy. To avoid this, we opted for a strategy that amalgamates the contextual information from these adj. and adv. words, rather than retaining nouns, verbs, adjectives, and adverbs simultaneously.

### 3.5 MULTI-GRAINED TEXTUAL-VISUAL ALIGNMENT

Given the multi-grained features $\{t^g, \{t^o\}, \{t^e\}\}$ and $\{v^g, \{v^o\}, \{v^e\}\}$ for all text-video pairs $\{(t, v)\}$, we calculate the overall cross-modal similarity $S$ for retrieval task by integrating similarities across multiple granularities according to Eq. 3.

**Global alignment.** Global features are employed to compare the similarity at the video-sentence level. Specifically, we use cosine similarity to measure the match between global text feature $\boldsymbol{t}^g$ and global video feature $\boldsymbol{v}^g$ of the two modalities, which is defined as:

$$\boldsymbol{S}^g = \frac{(\boldsymbol{t}^g)^{\text{T}} \boldsymbol{v}^g}{\|\boldsymbol{t}^g\| \, \|\boldsymbol{v}^g\|}. \tag{11}$$

Since global similarities between query and easy negative samples are much lower, most negative samples can be identified with global similarity alone. Therefore, we select only the $H$ most similar samples with the highest global similarity for further fine-grained object and event alignment. This coarse filtering substantially improves the computational efficiency of the proposed method.

The computation of global similarity is both low-cost and efficient. Yet, it struggles with capturing fine-grained details, leading to inferior accuracy and challenges in distinguishing between positive and hard samples.

**Object alignment.** Then we capture the static object-level similarity between the text and video. Since an object may appear across multiple frames in the video and not every frame in a positive sample contains the object mentioned in the text query, we select the top-$K$ the most similar video objects $\{v^o\}$ with text object $t^o$ as the matching. Additionally, to address the scenario where negative samples might partially match the text query, we incorporate a logarithmic function to amplify the disparity in unmatched pairs. This process can be formulated as:

$$S^o = \frac{1}{N_{noun}} \sum_{i=1}^{N_{noun}} \sum_{v_j^o \in \text{KNN}(t_i^o, K)} \log\left(\frac{(t_i^o)^{\mathrm{T}} v_j^o}{\|t_i^o\| \|v_j^o\|}\right), \tag{12}$$

where we define $\text{KNN}(t_i^o, K)$ as the $K$-nearest neighbors of text object $t_i^o$ from the video object $\{v^o\}$ according to cosine similarity, $N_{noun}$ is the number of text object tokens.

**Event alignment.** Given that there are also partial matches between events in videos and text, we employ a similar method to calculate event similarity, which can be written as:

$$S^e = \frac{1}{N_{verb}} \sum_{i=1}^{N_{verb}} \sum_{v_j^e \in \text{KNN}(t_i^e, K)} \log\left(\frac{(t_i^e)^{\mathrm{T}} v_j^e}{\|t_i^e\| \|v_j^e\|}\right). \tag{13}$$

In the end, Eq. 3 calculate the final similarity $S$ for the text-video pair $(t, v)$ by summing the three levels of similarity.

**Training Objective.** We treat text-video pairs as positive examples and treat all other combinations in the batch as negatives. For a batch consisting of $B$ pairs of (video, text), the model generates and optimizes $B \times B$ similarity scores. We employ symmetric cross-entropy loss on these scores for the model training:

$$\mathcal{L}_{v2t} = -\frac{1}{B} \sum_v \log\left(\frac{\exp(S_+)}{\exp(S_+) + \sum_t \exp(S_-)}\right), \tag{14}$$

$$\mathcal{L}_{t2v} = -\frac{1}{B} \sum_t \log\left(\frac{\exp(S_+)}{\exp(S_+) + \sum_v \exp(S_-)}\right), \tag{15}$$

where $S_+$ and $S_-$ denote similarities between the positive and negative text-video pairs, separately. The final loss function, $\mathcal{L}$, is a composite of the video-to-text loss, $\mathcal{L}_{v2t}$ and the text-to-video loss, $\mathcal{L}_{t2v}$, which can be written as:

$$\mathcal{L} = \mathcal{L}_{v2t} + \mathcal{L}_{t2v}. \tag{16}$$

## 4 EXPERIMENTS

We conduct experiments on three popular text-video retrieval datasets, i.e., MSRVTT (Xu et al., 2016), DiDeMo (Anne Hendricks et al., 2017) and MSVD (Wu et al., 2017). More details about dataset, evaluation metrics, and implementation can be seen in appendix A.

### 4.1 COMPARISON WITH STATE-OF-THE-ART METHODS

In Table 1 and Table 2, we compare the VideoUntier with recent works on the 9k and 7k splits of MSRVTT (Xu et al., 2016), respectively. BLIP (Li et al., 2022) only reports the zero-shot performance. VideoUntier outperforms other fine-grained alignment-focused models like HBI (Jin et al., 2023a), TS2-Net (Liu et al., 2022), and ProST (Li et al., 2023), and UCOFIA (Wang et al., 2023), which integrates both fine and coarse-grained alignments, in both text-to-video and video-to-text retrieval performance. Notably, it shows improvements of +0.8%, +2.4%, and +1.2% in text-to-video R@1 metric over HBI (Jin et al., 2023a), TS2-Net (Liu et al., 2022), and ProST (Li et al., 2023), respectively, and a +1.4% improvement over UCOFIA (Wang et al., 2023) in video-to-text R@1 metric. This result verifies our motivation that language-guided video feature extraction better aligns the cross-modal information and bolsters retrieval performance. When replacing the backbone to CLIP (VIT-B/16) (Radford et al., 2021), our method boosts the text-to-video and video-to-text R@1 metrics by 2.1% and 2.9%, respectively, over ProST (Li et al., 2023), achieving state-of-the-art.

Table 1: Comparisons to current works on the MSRVTT-9k (Xu et al., 2016) dataset. TVMM* (Lin et al., 2022) performance is reproduced using backbone network CLIP (ViT-B/32) (Radford et al., 2021). "†" denotes the model with CLIP (ViT-B/16) as backbone. The inference time is measured on a single NVIDIA RTX 3090 GPU.

| Method | Text → Video | | | | Video → Text | | | | Inference |
|---|---|---|---|---|---|---|---|---|---|
| | R@1↑ | R@5↑ | R@10↑ | MnR↓ | R@1↑ | R@5↑ | R@10↑ | MnR↓ | Time↓ |
| CLIP4Clip | 44.5 | 71.4 | 81.6 | 15.3 | 42.7 | 70.9 | 80.6 | 11.6 | 16.1s |
| CenterCLIP | 44.2 | 71.6 | 82.1 | 15.1 | 42.8 | 71.7 | 82.2 | 10.9 | - |
| X-Pool | 46.9 | 72.8 | 82.2 | 14.3 | - | - | - | - | - |
| TVMM* | 45.8 | 71.7 | 81.9 | 14.8 | 44.0 | 71.9 | 82.3 | 10.6 | 26.2s |
| TS2-Net | 47.0 | 74.2 | 83.3 | 13.6 | 44.3 | 73.9 | 83.0 | 9.2 | 19.8s |
| VoPF | 44.6 | 69.9 | 80.3 | 16.3 | 44.5 | 70.7 | 80.6 | 11.5 | - |
| HBI | 48.6 | 74.6 | 83.4 | 12.0 | 46.8 | 74.3 | 84.3 | 8.9 | - |
| DiffusionRet | 49.0 | 75.2 | 82.7 | 12.1 | 47.7 | 73.8 | 84.5 | 8.8 | 60.7s |
| UCOFIA | 49.4 | 72.1 | - | 12.9 | 47.1 | 74.3 | - | - | 23.5s |
| ProST | 48.2 | 74.6 | 83.4 | 12.4 | 46.3 | 74.2 | 83.2 | 8.7 | 25.2s |
| ProST† | 49.5 | 75.0 | 84.0 | 11.7 | 48.0 | 75.9 | 85.2 | 8.3 | 43.4s |
| VideoUntier | 49.4 | 75.1 | 83.5 | 11.7 | 48.5 | 74.4 | 84.5 | 8.2 | 20.1s |
| VideoUntier † | **51.6** | **78.4** | **85.1** | **9.8** | **50.9** | **76.7** | **85.4** | **7.6** | 38.8s |

Table 2: Text-to-Video retrieval results on the MSRVTT-7k (Xu et al., 2016) dataset.

| Method | Text → Video | | | |
|---|---|---|---|---|
| | R@1↑ | R@5↑ | R@10↑ | MnR↓ |
| HowTo100M | 14.9 | 40.2 | 52.8 | - |
| ClipBERT | 22.0 | 46.8 | 59.9 | - |
| CLIP4Clip | 42.1 | 71.9 | 81.4 | 15.7 |
| X-Pool | 43.9 | 72.5 | 82.3 | 14.6 |
| TS2-Net | 43.1 | 72.2 | 82.1 | 14.2 |
| BLIP | 43.3 | 65.6 | 74.7 | - |
| ProST | 44.5 | 72.3 | 82.4 | 13.8 |
| VideoUntier | **45.9** | **73.2** | **82.5** | **13.1** |

Table 3: Text-to-Video retrieval results on the DiDeMo (Anne Hendricks et al., 2017) dataset.

| Method | Text → Video | | | |
|---|---|---|---|---|
| | R@1↑ | R@5↑ | R@10↑ | MnR↓ |
| Frozen | 31.0 | 59.8 | 72.4 | - |
| TVMM | 36.5 | 64.9 | 75.4 | - |
| TS2-Net | 41.8 | 71.6 | 82.0 | 14.8 |
| CLIP4Clip | 42.8 | 68.5 | 79.2 | 18.9 |
| ProST | 44.9 | 72.7 | 82.7 | 13.7 |
| HBI | 46.9 | 74.9 | 82.7 | 12.1 |
| UCOFIA | 46.5 | 74.8 | - | 13.4 |
| VideoUntier | **47.5** | **75.2** | **82.9** | **11.9** |

Regarding computational efficiency, we compared the inference time with recent works. As shown in Table 1, our method maintains high computational efficiency with coarse strategy, achieving better performance (49.4% vs. 48.2%) in less inference time (20.1s vs. 25.2s with ProST (Li et al., 2023)).

In Table 3, 4 and 5, we assess the performance on DiDeMo text-to-video, on DiDeMo video-to-text and MSVD text-to-video tasks, respectively. Compared with recent ProST (Li et al., 2023), HBI (Jin et al., 2023a) and UCOFIA (Wang et al., 2023), our method achieves a boost of 2.6%, 0.6%, and 1.0% in R@1 metric on DiDeMo text-to-video, respectively. Across these tasks, our work consistently exceeds current state-of-the-art techniques, demonstrating the versatility and strong generalization of VideoUntier across various video domains.

## 4.2 ANALYSIS

**Robustness of the learned representation.** The robustness of learnt features can be improved with the proposed VideoUntier. Table 6 presents the effectiveness of our method in domain generalization against recent works. We pre-train a model on a source dataset and assess its performance on a different target dataset without fine-tuning. The results demonstrate that our method more effectively transfers knowledge from the source domain to new domains, surpassing recent works specializing in domain generalization (Jin et al., 2023b) by 2.6% and 2.2% in the R@1 metric on the DiDeMo and MSVD datasets, respectively.

**Effect of multi-grained similarities.** To achieve more intricate cross-modal alignment, we utilized similarity measures at three levels—global, object, and event—to collectively determine the overall similarity score. In Table 7, model trained with multi-similarities is tested using its individual similarity. With the same individual global features used in inference, the proposed method surpasses CLIP4Clip (Luo et al., 2022) by +0.7% in R@1. This shows that the proposed language-guided method not only provides a granular alignment way, but also enables the backbone to learn better feature representations. Incorporating object-level similarity enabled our model to effectively align

Table 4: Video-to-Text retrieval results on the DiDeMo (Anne Hendricks et al., 2017) dataset.

Table 5: Text-to-Video retrieval results on the MSVD (Wu et al., 2017) dataset.

| Method | Video → Text | | | |
|---|---|---|---|---|
| | R@1↑ | R@5↑ | R@10↑ | MnR↓ |
| FSE | 13.1 | 33.9 | - | - |
| S2V | 13.2 | 33.6 | - | - |
| CE | 15.6 | 40.9 | - | 42.4 |
| TT-CE | 21.1 | 47.3 | 61.1 | - |
| CLIP4Clip | 41.4 | 68.2 | 79.1 | 12.4 |
| HBI | 46.2 | 73.0 | **82.7** | 8.7 |
| UCOFIA | 46.0 | 71.9 | - | - |
| VideoUntier | **46.8** | **73.2** | 81.3 | **8.5** |

| Method | Text → Video | | | |
|---|---|---|---|---|
| | R@1↑ | R@5↑ | R@10↑ | MnR↓ |
| CE | 19.8 | 49.0 | 63.8 | - |
| SUPPORT | 28.4 | 60.0 | 72.9 | - |
| CLIP | 37.0 | 64.1 | 73.8 | - |
| Frozen | 33.7 | 64.7 | 76.3 | - |
| TVMM | 36.7 | 67.4 | 81.3 | - |
| CLIP4Clip | 45.2 | 75.5 | 84.3 | 10.3 |
| X-Pool | 47.2 | 77.4 | 86.0 | 9.3 |
| VideoUntier | **48.5** | **78.2** | **86.5** | **8.6** |

Table 6: Domain generalization performance. The "A->B" signifies that "A" represents the source domain, while "B" denotes the target domain. CLIP4Clip* (Luo et al., 2022) performance is reproduced by our own.

| Method | MSRVTT | | | | MSRVTT->DiDeMo | | | | MSRVTT->MSVD | | | |
|---|---|---|---|---|---|---|---|---|---|---|---|---|
| | R@1↑ | R@5↑ | R@10↑ | MdR↓ | R@1↑ | R@5↑ | R@10↑ | MdR↓ | R@1↑ | R@5↑ | R@10↑ | MdR↓ |
| CLIP4Clip* | 43.8 | 70.6 | 81.4 | 2.0 | 31.8 | 57.0 | 66.1 | 4.0 | 15.3 | 31.3 | 40.5 | 21.0 |
| EMCL-Ne | 47.0 | 72.3 | 82.6 | 2.0 | 30.0 | 56.1 | 65.8 | 4.0 | 16.6 | 29.3 | 36.5 | 24.0 |
| DiffusionRet | 49.0 | 75.2 | 82.7 | 2.0 | 33.2 | 59.3 | 68.4 | 3.0 | 17.1 | 32.4 | 41.0 | 21.0 |
| VideoUntier | **49.4** | **75.1** | **83.5** | **2.0** | **35.8** | **62.6** | **69.6** | **3.0** | **19.3** | **34.5** | **41.7** | **20.0** |

static object information across modalities with fine granularity, increasing the R@1 accuracy by +2.9%. Employing all three levels of similarity, the proposed method improved the R@1 accuracy by +4.2% compared to global features alone, validating the effectiveness of aligning cross-modal features at multiple granularities.

**Effect of the number of hard samples.** To enhance the computational efficiency, we introduced a coarse filter strategy. This involves using global coarse features to select $H$ hard samples for further fine-grained comparison. In Table 8, we assess the effect of varying $H$ on both performance and efficiency. The 2-nd and the 6-th rows of the table indicate that using all samples for fine-grained calculation yields a small performance rise of only 0.2% (49.6% for R@1), but at a computational time ×11.3 times longer than the coarse filter approach (227.01s vs. 20.07s).The results verify that our coarse filter strategy effectively improves computational efficiency without severe performance loss.

**Effect of number of top-$K$ matching.** In case some negative sample videos partially align with text semantics, and positive samples may not fully reflect these semantics in all frames, we introduce a strategy to use the top-$K$ most similar text-video pairs for fine-grained similarities. As shown in Table 8, $K = N_f/2$ surpasses $K = N_f$ by 0.8% on R@1, indicating that some objects do not appear in every frame in positive video frames and proving the effectiveness of the top-$K$ strategy.

**Visualization.** In Figure 3, we show the attended video regions of global feature and the extracted object and event features. Query-agnostic global-level feature attends to various contents of video, including objects in noisy background. It can be seen that the proposed method effectively identifies visual features related to objects and events in the query. Such features reinforce the precise alignment between video contents and textual semantics. For example, the visual features extracted based on the word 'bus' can accurately focus on the bus area in the first two frames, and not interfere by the next two frames. Figure 3 also shows that relying solely on global-level features fails to mine valuable clues in the video. Our retrieval pipeline thus employs global features to quickly narrow down the search scope, hence conducting retrieval with object-level and event-level features to ensure high accuracy. See the appendix C for more visualizations.

## 5 CONCLUSION

In this work, we focus on the challenge of inherent modal heterogeneity in video feature learning. Particularly, in Text-Video Retrieval (TVR) task, videos always include much query-irrelevant noise,

Table 7: Ablation study on the effect of multi-grained similarities, i.e. global similarity $\boldsymbol{S}^g$, object similarity $\boldsymbol{S}^o$ and event similarity $\boldsymbol{S}^e$.

| $\boldsymbol{S}^g$ | $\boldsymbol{S}^o$ | $\boldsymbol{S}^e$ | Text → Video | | | |
|---|---|---|---|---|---|---|
| | | | R@1↑ | R@5↑ | R@10↑ | MnR↓ |
| ✓ | | | 45.2 | 71.7 | **83.8** | 13.4 |
| ✓ | ✓ | | 48.3 | 74.1 | 83.2 | 12.2 |
| ✓ | | ✓ | 47.1 | 73.0 | 82.3 | 13.6 |
| ✓ | ✓ | ✓ | **49.4** | **75.1** | 83.5 | **11.7** |

Table 8: Ablation study about the number of hard samples $H$ and $\boldsymbol{K}$ for Top-K similarity matching.

| $H$ | $K$ | Text → Video | | | | Inference Time↓ |
|---|---|---|---|---|---|---|
| | | R@1↑ | R@5↑ | R@10↑ | MnR↓ | |
| - | - | 47.2 | 74.3 | 83.3 | 12.5 | 227.0s |
| - | $N_f/2$ | **49.6** | **75.6** | **84.3** | **11.2** | 227.1s |
| 20 | $N_f$ | 48.8 | 74.5 | 83.7 | 11.9 | 25.67s |
| 20 | $N_f/2$ | 49.5 | 75.2 | 84.0 | 11.6 | 25.38s |
| 10 | $N_f$ | 48.6 | 74.2 | 83.5 | 12.1 | 20.11s |
| 10 | $N_f/2$ | 49.4 | 75.1 | 83.5 | 11.7 | **20.07s** |

Figure 3: Visualization of global features, event tokens, and object tokens extracted from the video. Features similar to the global feature or illustrated words are highlighted.

hindering accurate textual-visual alignment. To precisely extract visual features that align with text queries, we propose a novel VideoUntier framework. The framework employs a Part-of-Speech-based Token Generator (PTG) module to extract object and event tokens from the text query. Subsequently, the Language-Guided Progressive Merging (LGPM) module leverages these query tokens as a guide to aggregate corresponding visual object and event features from the video. Experiments on three TVR datasets show that VideoUntier achieves precise fine-grained alignment across different modalities, enhancing the robustness of the learnt features. To the best of our knowledge, this work is an original effort in learning object and event features from videos by the guidance from text queries in TVR, which may inspire future work on text-guided video feature learning.

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

## A  EXPERIMENTAL SETTINGS

**Datasets.** We conduct experiments on three popular text-video retrieval datasets, i.e., MSRVTT (Xu et al., 2016), DiDeMo (Anne Hendricks et al., 2017) and MSVD (Wu et al., 2017). Data Pre-processing follows common practice (Luo et al., 2022; Jin et al., 2023a) for fair comparison. **MSRVTT** (Xu et al., 2016) comprises 10K videos paired with 200K human-generated captions. For thoroughly comparison with existing methods, we follow (Lin et al., 2022; Li et al., 2023) training settings, utilizing either 7K or 9K videos from the combined training and validation sets, and evaluating on a distinct test set of 1K text-video pairs. **DiDeMo** (Anne Hendricks et al., 2017) consists of 10K videos annotated with 40,000 sentences. Following the common practice (Jin et al., 2023a; Wang et al., 2023), sentences for each video are concatenated to form a single query for text-video retrieval. There are 8K videos in the training set, ∼1K videos in the validation set, and ∼1K videos in the test set. **MSVD** (Wu et al., 2017) contains 1.9K videos, varying in duration from one to 62 seconds. Train, validation and, test splits contain 1,200, 100, and 670 videos, respectively, with an average of 40 English sentences describing each video.

**Evaluation Metrics.** We abbreviate Recall at K to R@K (K = 1, 5, 10) upon all datasets. The MdR represents the median position of the ground truth within the retrieval sequence, whereas the MnR reflects the average ranking of accurate results. Note that higher R@K (indicated as ↑) and lower MdR and MnR (indicated as ↓) mean better performance.

**Implementation Details.** Following previous works (Luo et al., 2022; Wang et al., 2023; Li et al., 2023), we employ the pre-trained CLIP (ViT-B/32 (Radford et al., 2021)) weights to initialize our model. The dimension $d$ of visual and textual representations is set to 512. Input videos are resized to $224 \times 224$ with random cropping and horizontal flipping. In alignment with existing practice (Luo et al., 2022; Jin et al., 2023a), we set the sampled frame number as 12 for MSRVTT (Xu et al., 2016) and MSVD (Wu et al., 2017) datasets, and max text query length at 24. For the paragraph-to-video datasets DiDeMo (Anne Hendricks et al., 2017), frame sampling number increases to 64 frames per video, with text queries limited to a length of 64. The Adam optimizer (Hu et al., 2022) is utilized, complemented by a cosine warm-up strategy (Loshchilov & Hutter, 2016). The learning rate is set to 1e-7 for CLIP-initialized weights and 1e-4 for all other parameters. The batch size is set to 128 for MSRVTT and MSVD datasets, and 64 for Didemo. The coarse-to-fine strategy is applied in both training and testing time. Global similarity is computed in all text-video pairs and the object and event similarities are only computed in the $H$ hard pairs for each query, where $H$ is set to 10. In Eq. 12 and Eq. 13, the neighbor size $K$ is set to half of frame numbers $N_f$ ($K$ is 6 for MSRVTT). Ablation studies are conducted on the most popular MSRVTT dataset to analyze the effect of different designs of our model.

## B  PoS TAGGING

In the Part-of-Speech-based Token Generator (PTG) module, we employ part-of-speech (PoS) tagging to parse sentences. Specifically, we use the off-the-shelf Stanford PoS tagger (Manning et al., 2014), implemented in the Natural Language Toolkit (NLTK) (Loper & Bird, 2002), to assign tags such as 'nouns' and 'verbs' to each word in a sentence. This toolkit encompasses key NLP techniques like sequence labeling, n-gram models and backoff, which are vital across various applications. In NLTK, a tagged token is typically represented as a tuple, comprising the token and its corresponding tag. For example, the sentence "boys play basketball" is annotated using the Penn Treebank tagset as [('boys','NNS'), ('play','VB'), ('basketball','NN')], where 'NNS' denotes plural nouns, and 'VB' and 'NN' represent the verb and nouns in base form, respectively. Words unrecognized by the tagger during its training are assigned a tag of 'None'. Based on these tags, words like 'boys' and 'basketball' are used for object feature extraction as nouns, and 'play' as a verb for event feature extraction.

## C  VISUAL ANALYSIS

For inconsistency between textual queries and video information density, this work introduces a language-guided approach for extracting video features, hence facilitating precise alignment between textual and visual features. This section visualizes how text and visual features correspond across

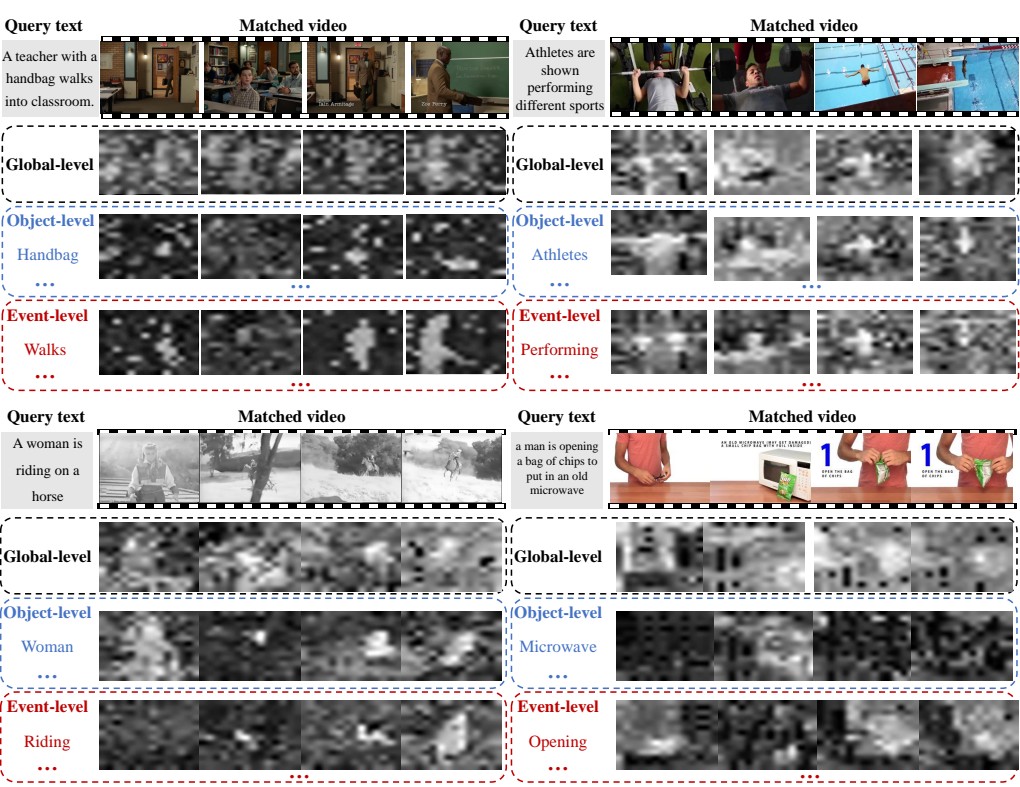

Figure 4: Visualization of multi-granularity alignment. The first row displays the query text alongside the corresponding matched video clip. The subsequent three rows show the video regions matched with global features, object tokens, and event tokens, respectively.

global, object, and event levels. For global features, we highlight video patches that are similar to global visual features. For object and event features, we display the nouns and verbs from the sentences, along with the visual feature regions aggregated under their guidance. As illustrated in Figure 4, global features reflect the global information of video frames but may not focus on the region of interest of the text query. In contrast, object and event features extracted under textual guidance align well with key contents in the text query. For example, in the bottom right example, global features are distracted by caption and numbers shown in the video frames. However, under the guidance of the noun "microwave", the visual features concentrate on the microwave appearing in the second frame. Similarly, under the guidance of the verb "open", the extracted event-level features focus on the person performing the action and the act of opening a bag. This brings more precise video feature extraction, aligning accurately with the text at multiple granularities, thereby enhancing retrieval performance.

