# OpenReview forum: "VideoUntier: Language-guided Video Feature Disentanglement"
_ICLR.cc/2025/Conference — ICLR 2025 Conference Withdrawn Submission_

### Official Review · Reviewer_WoJi · 2024-10-31

**Soundness:** 3
**Presentation:** 3
**Contribution:** 2
**Rating:** 5
**Confidence:** 3

**Summary:**

The paper addresses the challenge of learning spatio-temporal representations for video understanding, especially in the context of Text-Video Retrieval (TVR). Traditional methods struggle with aligning video and text features due to the noise from irrelevant visual cues. The authors propose VideoUntier, a method that disentangles video content into global, object, and event levels using text query guidance. This approach aims to improve the alignment between textual and visual features by generating compact and meaningful video features. Experimental results demonstrate that VideoUntier outperforms existing methods in accuracy and efficiency, offering robust feature alignment and potential for future research advancements in video-text retrieval.

**Strengths:**

1. The concept of decomposing query tokens into object and event tokens is intriguing, as it aids the model in more accurately identifying various aspects of the video content.

2. The ablation study effectively highlights the benefits of separating objects and events.

**Weaknesses:**

1. The authors overlook a crucial performance comparison with recent works such as CLIP-VIP [1] and T-Mass [2], which are essential for assessing the proposed method’s superiority.

2. While decomposing the text query into object and event tokens can direct the model’s attention to specific video aspects, it might compromise the query’s coherence and completeness. I recommend conducting an experiment that uses the entire sequence of tokens in the PTG module without separating them into nouns and verbs, followed by applying the LPVM module to model temporal relations. This experiment will provide insights into the effectiveness of the decomposition.


[1] Xue, Hongwei, et al. "Clip-vip: Adapting pre-trained image-text model to video-language representation alignment." arXiv preprint arXiv:2209.06430 (2022).

[2] Wang, Jiamian, et al. "Text Is MASS: Modeling as Stochastic Embedding for Text-Video Retrieval." Proceedings of the IEEE/CVF Conference on Computer Vision and Pattern Recognition. 2024.

**Questions:**

1. Add the missing literature, e.g., CLIP-VIP and T-Mass.

2. Conduct an additional experiment to further explain the effectiveness of the decomposition design.

---

### Official Review · Reviewer_x1Sm · 2024-11-02

**Soundness:** 3
**Presentation:** 3
**Contribution:** 2
**Rating:** 5
**Confidence:** 4

**Summary:**

The paper proposes VideoUntier, a new method for text-video retrieval that decompose video features into global-, object- and event-level to improve alignment with textual cues. By generating object and event tokens from text queries to guide feature extraction, VideoUntier creates compact visual representations that outperform existing methods.

**Strengths:**

1. Figures 1 and 2 effectively elucidate the core concepts and methodology of the research.
2. The paper is articulate and easy to comprehend.
3. The manuscript includes extensive discussions spanning related work to methodologies, offering deeper insights into the proposed approach.

**Weaknesses:**

1. Decomposing holistic visual cues into global-level, object-level and event-level is a known concept in the realm of text-video retrieval. Given the truth that the authors anchor object-level visual cues with nouns and event-level visual cues with verbs in the corresponding text using an attention-based method, this approach has also been explored in the literature. Some prior works delve even further, utilizing optical flow or other motion-based information to model event-level cues.

Some closely related works that share similar ideas:

[1] Text-Video Retrieval with Disentangled Conceptualization and Set-to-Set Alignment. In IJCAI, 2023.

[2] Beyond Coarse-Grained Matching in Video-Text Retrieval. In ACCV, 2024.

[3] Coarse-to-fine dual-level attention for video-text cross modal retrieval. KBS, 2022.

[4] Fine-grained Video-Text Retrieval with Hierarchical Graph Reasoning. In CVPR, 2020.

2. On the other hand, while the hierarchical modeling (global-and local-level) for text-video retrieval is a commendable concept, given the existing efforts that share similar knowledge within the community, the novelty this work brings might be perceived as marginal for publication in ICLR.

Some closely related works that share similar ideas:

[1]T2VLAD: Global-Local Sequence Alignment for Text-Video Retrieval. In CVPR, 2021.

[2]Fine-grained Video-Text Retrieval with Hierarchical Graph Reasoning. In CVPR, 2020.

[3]Cross-Modal and Hierarchical Modeling of Video and Text. In ECCV, 2018.

3. I may suggest that the authors reconsider the term 'DISENTANGLEMENT' and opt for 'decomposition' instead, as object-level information and event-level information are inherently interconnected, with some events directly involving specific objects.

**Questions:**

One typo: Ln046 - fig:teaser fails to link to the figure.

---

### Official Review · Reviewer_VkyC · 2024-11-03

**Soundness:** 3
**Presentation:** 3
**Contribution:** 2
**Rating:** 1
**Confidence:** 5

**Summary:**

The proposed VideoUntier framework is introduced as an innovative approach for Text-Video Retrieval to address the alignment challenges between textual queries and complex video content. By utilizing a Part-of-Speech-based Token Generator (PTG) and a Language-Guided Progressive Vision Merging (LGPM) module, the proposed method extracts relevant object and event features from videos guided by text, leading to more precise multi-granular feature alignment. This framework outperforms existing models in terms of retrieval performance on datasets like MSRVTT, MSVD, and DiDeMo.

**Strengths:**

1. The proposed method include global, object, and motion alignment is novel.
2. The proposed PTG and LGPM are interesting.
3. The performance is good.

**Weaknesses:**

1. Efficiency.
	1. Video object features rely on the object-level feature from text. The training and test efficiency is low as this model is a single tower model. It needs O(mn) complexity if m and n are the number of video and text.
2. While visual object features rely on text feature, the visual motion features are not. What is the performance of also using text motion features?
3. On the other hand, what is the performance of not using text features at all for calculating visual features?
4. Lack of ablations on LGPM and PTG modules.
5. Lack of 2024 papers, such as Textismass [1].
6. The main paper is over 9 pages. Should be desk rejected.

[1] Text Is MASS: Modeling as Stochastic Embedding for Text-Video Retrieval. CVPR 24.

**Questions:**

See weakness

---

### Official Review · Reviewer_Cij2 · 2024-11-04

**Soundness:** 2
**Presentation:** 3
**Contribution:** 2
**Rating:** 3
**Confidence:** 5

**Summary:**

This paper introduces a method to improve text-video retrieval (TVR) by addressing the misalignment between complex video features and concise text queries. The proposed framework, VideoUntier, leverages object and event tokens extracted from text queries to guide video feature extraction, reducing noise and enhancing alignment. It consists of two main components: the Part-of-Speech-based Token Generator (PTG) and the Language-guided Progressive Vision Merging (LPVM) module. Experiments on MSRVTT, MSVD, and DiDeMo datasets show that VideoUntier outperforms state-of-the-art methods in accuracy and efficiency, demonstrating the benefits of text-guided video feature disentanglement.

**Strengths:**

- The paper is clearly written and structured, with detailed explanations of the proposed framework and well-designed diagrams that illustrate the process of language-guided feature disentanglement.
- The work is well-executed, with comprehensive experiments demonstrating that VideoUntier consistently outperforms state-of-the-art approaches across multiple benchmarks, such as MSRVTT, MSVD, and DiDeMo.

**Weaknesses:**

- The main concern is the novelty. The core idea of using object and event text features to guide video feature extraction has similarities to existing works, such as "Fine-grained video-text retrieval with hierarchical graph reasoning" (CVPR 2020). This work should be cited and compared to, as the proposed cross-attention-based object merger is conceptually similar to the attentive matching techniques previously explored.
- The experimental section lacks evaluations against some recent and relevant methods. Specifically, baseline comparisons with works like "Cap4Video: What Can Auxiliary Captions Do for Text-Video Retrieval?" (CVPR 2023), "T2VLAD: Global-Local Sequence Alignment for Text-Video Retrieval" (CVPR 2021), and "Disentangled Representation Learning for Text-Video Retrieval" (ECCV 2022) are essential for a comprehensive assessment of the method’s effectiveness. The omission of these baselines reduces the clarity of how the proposed model stands relative to state-of-the-art approaches.
- The reliance on a Part-of-Speech-based Token Generator for parsing sentences introduces additional computational overhead. This design choice raises concerns about the model's practicality and scalability in real-world applications, where processing speed is critical.
- Given the complexity of the proposed framework, the performance improvements over existing methods are relatively modest.
- The parsing-based approach might not offer significant advantages over simpler models, such as DRL or X-Pool, which use direct word query-based matching.

**Questions:**

- Can the authors further differentiate the proposed method from similar existing works, such as "Fine-grained video-text retrieval with hierarchical graph reasoning" (CVPR 2020) and other cross-attention-based retrieval techniques?
- Given the modest performance improvement relative to the complexity of the proposed method, could the authors discuss potential use cases where the gains justify the added design complexity?
- How does the proposed framework handle more complex or ambiguous text queries that may not clearly separate objects and events?

---

### Note · Authors · 2024-12-02

I have read and agree with the venue's withdrawal policy on behalf of myself and my co-authors.